# Stability of Quantum-Dot Light Emitting Diodes with Alkali Metal Carbonates Blending in Mg Doped ZnO Electron Transport Layer

**DOI:** 10.3390/nano10122423

**Published:** 2020-12-04

**Authors:** Hyo-Min Kim, Wonkyeong Jeong, Joo Hyun Kim, Jin Jang

**Affiliations:** Department of Information Display and Advanced Display Research Center, Kyung Hee University, 26 Kyungheedae-ro, Dongdaemun-gu, Seoul 02447, Korea; hmkim@tft.khu.ac.kr (H.-M.K.); wkjeong@tft.khu.ac.kr (W.J.); jhkim3@tft.khu.ac.kr (J.H.K.)

**Keywords:** alkali metal carbonate, metal oxide, Mg doped ZnO, operational lifetime, QLED

## Abstract

We report here the fabrication of highly efficient and long-lasting quantum-dot light emitting diodes (QLEDs) by blending various alkali metal carbonate in magnesium (Mg) doped zinc oxide (ZnO) (MZO) electron transport layer (ETL). Alkali metal carbonates blending in MZO, X_2_CO_3_:MZO, control the band-gap, electrical properties, and thermal stability. This can therefore enhance the operational lifetime of QLEDs. It is found that the conductivity of X_2_CO_3_:MZO film can be controlled and the thermal stability of ETLs could be improved by X_2_CO_3_ blending in MZO. The inverted red QLEDs (R-QLEDs) with Cs_2_CO_3_:MZO, Rb_2_CO_3_:MZO, and K_2_CO_3_:MZO ETLs exhibited the operational lifetime of 407 h for the R-QLEDs with Cs_2_CO_3_:MZO, 620 h with Rb_2_CO_3_:MZO and 94 h with K_2_CO_3_:MZO ETLs at T_95_ with the initial luminance of 1000 cd/m^2^. Note that all red QLEDs showed the high brightness over 150,000 cd/m^2^. But the R-QLEDs with Na_2_CO_3_:MZO and Li_2_CO_3_:MZO ETLs exhibited shorter operational lifetime and poor brightness than the R-QLED with pristine MZO ETL.

## 1. Introduction

Colloidal quantum-dots (QDs) based light-emitting diodes (QLEDs) have been developed for next generation display replacing organic LEDs (OLEDs) because of the advantages of QDs such as easier color tuning, good color purity with narrow full-width at half maximum (FWHM), long lifetime with inorganic components, and facile synthetic technology [1,2,3,4]. The QLED efficiency is approaching that of OLED [5,6,7,8]. However, some challenges remain for commercialization, for example, the stability of solution processable materials, the relationship between efficiency roll-off and lifetime, and so on.

For the commercial applications of active-matrix (AM) displays, the inverted QLED is favored because the drain contact of n-type amorphous indium-gallium-zinc-oxide (a-IGZO) thin-film transistors (TFTs) is connected to a bottom cathode [9,10,11]. The solution processable n-type inorganic metal oxides, such as zinc oxide (ZnO), titanium oxide (TiO_x_), and tin oxide (SnO) with high stability can be used on the top of bottom cathode with organic pixel define layers (PDLs) [12,13,14]. Besides, the other charge transporting layers (CTLs) with hole transporting and injection layers (HTL and HIL) can be deposited on top of QD emissive layer (EML) by vacuum evaporation to avoid solution inter-mixing when full solution processing is used [15,16,17]. In this hybrid processed (solution process combined with thermal evaporation) QLEDs, the material stabilities, such as electrical and thermal stabilities, are one of key factors compared with thermal evaporation to obtain devices with high performance and a long lifetime.

For the efficient electron transport into organic or QD EMLs, an introduction of ultra-thin alkali metal carbonate (X_2_CO_3_, X = Li, Na, K, Rb and Cs) interlayers between electron injection layer (EIL) and EML has been suggested [18,19,20]. In general, the alkali metal carbonates are deposited by vacuum process, and the cesium carbonate (Cs_2_CO_3_) is widely used as ab interlayer on top of ZnO EIL. Note that it can be used for solution process because of its good solubility in polar solvents such as ethanol, acetone, and methanol [21,22]. For an efficient electron injection from cathode, the ultra-thin X_2_CO_3_ interlayer (thinner than 10 nm) needs to be deposited in addition to ETL because of its insulating characteristics [23,24]. Compared to the thermal evaporation process, the thickness control is one of key issues in the solution process to form a uniform and ultra-thin layer. However, the uniform and easier thickness controllable EIL/electron transporting layer (ETL) can be achieved without an additional interlayer deposition process when the alkali metal carbonate is doped into metal oxide [25,26,27].

Chen et al. reported the inverted OLEDs (i-OLEDs) using alkali metal carbonates (except rubidium carbonate, Rb_2_CO_3_) doped ZnO EIL [28]. For the i-OLEDs, the fluorescence tris-(8-hydroxyquinoline) aluminum (Alq_3_) was used as EML and it was confirmed that alkali metal carbonate increases the electron mobility of ZnO and reduces energy barrier for efficient electron injection into EML. In the report, the maximum current efficiency (CE_max_) of i-OLED was 6.04 cd/A with potassium carbonate (K_2_CO_3_) doped ZnO EIL. Jeong et al. also reported perovskite solar cells with alkali metal carbonate interlayers (except rubidium carbonate, Rb_2_CO_3_) on ZnO nanoparticles (NPs) ETL [29]. The maximum power conversion efficiency (PCE_max_) of the perovskite solar cells was 14.1% when Cs_2_CO_3_ interlayer was deposited on ZnO NPs with the alkali metal carbonate as an inter-layer on ZnO NPs.

In this study, we report the blending effect of alkali metal carbonates (X_2_CO_3_, X = Li, Na, K, Rb and Cs) in Mg doped ZnO (MZO) ETL and focus on which X_2_CO_3_ is the most effective dopant in MZO to improve red QLED (R-QLEDs) lifetime. The inverted R-QLEDs are suggested to monitor the device performance and lifetime, and the thermalgravimetric analysis (TGA), ultraviolet photoelectron spectroscopy (UPS), time-resolved photoluminescence (TRPL), atomic force microscopy (AFM), conductivity, capacitance, and electrical stress analysis are characterized for X_2_CO_3_ blended MZO (X_2_CO_3_:MZO) ETL. Through thin-film analysis, we confirmed the following facts; (i) X_2_CO_3_ blending in metal oxide increases a glass transition temperature (T_g_) because of its less polarizing effect by singly charged positive ions, (ii) work-function (WF) of MZO become close to conduction band minimum (CBM) and (iii) conductivity increases when large atomic compound-carbonate is mixed in MZO (from Li to Cs), (iv) large energy barrier between CBM and WF of X_2_CO_3_:MZO induces electron accumulation at the interface of EIL/ETL, therefore, it degrades the lifetime, and (v) Cs_2_CO_3_ and Rb_2_CO_3_ are the most suitable alkali metal carbonate dopants in MZO ETL to improve R-QLED lifetime.

## 2. Materials and Methods

In this study, metal oxide solutions were synthesized using a sol-gel process [30,31]. A 0.5 M zinc acetate dihydrate (Zn(C_4_H_6_O_4_)_2_·2H_2_O, Sigma Aldrich, Seoul, Korea) precursor was dissolved in 2-methoxyethanol (C_3_H_8_O_2_, 2-ME, Sigma Aldrich, Seoul, Korea) with monoethanolamine (NH_2_CH_2_CH_2_OH, MEA, Sigma Aldrich, Seoul, Korea) as a stabilizer and the mole ratio of MEA was kept at 1:1 with the precursor. From the pristine ZnO solution, lithium acetate hydrate (CH_3_COOLi∙H_2_O, Sigma Aldrich, Seoul, Korea) and magnesium acetate tetrahydrate ((CH_3_COO)_2_Mg·4H_2_O, Sigma Aldrich, Seoul, Korea) precursors were added into the pristine ZnO solution to obtain lithium doped ZnO (LZO) and MZO solutions, respectively. Atomic percentages for Li and Mg precursors in ZnO solution were fixed as 10 at% for each LZO and MZO solutions, respectively. Finally, the solutions were refluxed at 50 °C for 6 h until a clear solution was obtained. For X_2_CO_3_:MZO solutions, cesium carbonate (Cs_2_CO_3_), rubidium carbonate (Rb_2_CO_3_), potassium carbonate (K_2_CO_3_), sodium carbonate (Na_2_CO_3_), and lithium carbonate (Li_2_CO_3_) were purchased from Sigma Aldrich, Seoul, Korea and the X_2_CO_3_ precursors were added into the prepared 10% MZO solution, and the mixed X_2_CO_3_:MZO solutions were stirred again at 400 rpm for 24 h. Here, atomic percentages for X_2_CO_3_ precursors in MZO solution were fixed at 4% for the X_2_CO_3_:MZO solutions [32].

For inverted red QLED (R-QLED), patterned indium-tin-oxide (ITO) substrate was cleaned using acetone, methanol, and isopropyl alcohol with sonication for 15 min, respectively. Subsequently, a 60 nm LZO thin-film was formed on ITO and annealed at 300 °C for 10 min in air for EIL. Then, 60 nm MZO and X_2_CO_3_:MZO thin-films for ETL were deposited on LZO EIL and annealed at 220 °C for 30 min in N_2_. For QD EML, CdSeZnS/ZnS red QDs (R-QDs) solution supplied from ZEUS, Gyeonggi-do, Korea, was spin-coated on ETL and annealed at 190 °C for 10 min in N_2_. As the HTL and HIL, small molecule based 10 nm thick Tris(4-carbazoyl-9-ylphenyl)amine (TCTA), 20 nm thick N,N′-Di(1-naphthyl)-N,N′-diphenyl-(1,1′-biphenyl)-4,4′-diamine (NPB) and 20 nm thick 1,4,5,8,9,11-Hexaazatriphenylenehexacarbonitrile (HAT-CN) were thermally evaporated on QD EML under 10^−7^ Torr [33,34]. First, the TCTA layer was used as electron blocking layer (EBL) and HTL due to the shallow lowest unoccupied molecular orbital (LUMO) and deep highest occupied molecular orbital (HOMO) levels and NPB/HAT-CN junction was used for efficient charge generation junction (CGJ) [35]. After EBL and CGJ depositions, 100 nm Al was thermally evaporated on top of HAT-CN for the anode. Finally, the QLED was encapsulated in a N_2_ filled glove box using glass.

An Agilent 4156C semiconductor parameter analyzer (Agilent, Santa Clara, CA, USA) was used to monitor the electrical characteristics of the electron-only devices (EODs). TGA analysis of MZO and X_2_CO_3_:MZO solutions were performed using SDT-Q600 (TA instruments, New Castle, DE, USA). The absorbance and PL of the R-QDs, MZO and X_2_CO_3_:MZO thin-films were measured with a Scinco S-4100 UV−visible spectrophotometer and Jasco FP-6500 spectrofluorometer, respectively. And TRPL results of R-QDs were obtained using C11367-14 (HAMAMATSU, Japan). The AFM analysis of ITO/MZO and ITO/X_2_CO_3_:MZO layers were performed using XE-100 (Park Systems, Gyeonggi-do, Korea). The UPS results of ITO, ITO/MZO, and ITO/X_2_CO_3_:MZO layers were obtained using Ulvac-PHI. Moreover, C-V results of EODs with various X_2_CO_3_:MZO ETLs were obtained using Agilent E4980A precision LCR meter. The current density−voltage (J-V) and luminance−voltage (L-V) characteristics were measured using a Konica Minolta CS-100A luminance meter coupled with a Keithley 2635A voltage and current source meter. Finally, the operational lifetime of R-QLEDs with X_2_CO_3_ ETLs was measured using M6000 (McScience, Gyeonggi-do, Korea).

## 3. Results and Discussion

### 3.1. Material Information of R-QDs

Appendix A exhibits the optical characteristics of CdZnSeS/ZnS R-QDs used in this study. The R-QDs was dissolved in octane with concentration of 10 mg/mL and diameter of ~10 nm, and the R-QD solution was supplied from ZEUS, Gyeonggi-do, Korea. To dissolve R-QDs in octane solvent, oleic acid (OA) and octanethiol (OT) were selected as ligands. The quantum-yield (QY) of R-QD solution was measured as 91%, with photoluminescence (PL) peak wavelength and FWHM of R-QDs of 620 nm and 27 nm, respectively, as shown in Appendix A. Appendix A exhibits the summarized chemical information of alkali metal carbonates.

### 3.2. Thin-Film Analysis for X_2_CO_3_:MZO ETLs

TGA is one of key factors to estimate thermal stability of material [36]. Through the TGA analysis, the annealing temperature or thermal stability characteristics of CTLs are estimated. Figure 1 exhibits the TGA result of X_2_CO_3_:MZO solutions, performed at N_2_ environment with a heating rate of 10 °C/min. The large solution weight loss up to 120 °C in Figure 1a, is due to the boiling point (b.p) of 2-methoxyethanol (~120 °C). The T_g_ of X_2_CO_3_:MZO solutions are confirmed as 249 °C for Cs_2_CO_3_:MZO, 259 °C for Rb_2_CO_3_:MZO, 251 °C for K_2_CO_3_:MZO, 236 °C for Na_2_CO_3_:MZO, and 245 °C for Li_2_CO_3_:MZO solutions as shown in Figure 1a. Figure 1b shows the summarized T_g_ characteristics of X_2_CO_3_:MZO solutions with a reference of a pristine MZO solution. It is noted that pristine MZO solution exhibited a low T_g_ of 218 °C in our previous work [32]. Compared to the pristine MZO solution, all X_2_CO_3_:MZO solutions exhibited a relatively higher T_g_, which affects the device degradation. These improved T_g_ by introducing X_2_CO_3_ in MZO can be explained as thermal decomposition [37]. In general, most carbonates tend to decompose by heating to form the metal oxide and carbon dioxide. During the decomposition process, the carbonate ion becomes polarized and the polarizing effect depends on positive ion [38]. Therefore, the Group 1 compounds (Li, Na, K, Rb, and Cs) having one positive charge exhibit a less polarizing effect than those of Group 2 compounds (Be, Mg, Ca, Sr and Ba). Therefore, the Group 1 compounds-based carbonates (X_2_CO_3_) have a thermally stable characteristic [39].

The energy level alignment of the CTLs supports a lot of electrical information such as WF, the energy barrier between EML and the charge injection layer, and the semiconductor type (n-type or p-type, strong or weak). Figure 2 exhibits UPS results and energy band diagram of X_2_CO_3_:MZO ETLs. It is noted that X_2_CO_3_:MZO ETLs are formed on ITO and its thickness is 60 nm, and He I (21.2 eV) ionization energy was used for UPS measurement. As shown in Figure 2a,b, the vacuum level and valance band shifts (ΔE_vac_ and ΔVB) for Cs_2_CO_3_:MZO, Rb_2_CO_3_:MZO, K_2_CO_3_:MZO, Na_2_CO_3_:MZO, and Li_2_CO_3_:MZO are found to be 2.12, 1.92, 1.63, 1.59 and 1.63 eV, and 3.37, 3.33, 3.21, 3.09, and 2.5 eV, respectively. The optical band-gaps (E_opt_) for Cs_2_CO_3_:MZO, Rb_2_CO_3_:MZO, K_2_CO_3_:MZO, Na_2_CO_3_:MZO, and Li_2_CO_3_:MZO are 3.40, 3.52, 3.49, 3.61 and 3.61 eV by the Tauc plot, respectively, as shown in Figure 2c. The pristine MZO ETL has 1.10 eV for ΔE_vac_, 3.37 eV for ΔVB and 3.67 eV for E_opt_ [32]. Figure 2d exhibits the energy band diagram of X_2_CO_3_:MZO resulted from UPS data and the details are summarized in Table 1. It is confirmed that the WF of X_2_CO_3_:MZO ETL decreases along with the decreasing atomic number compound alkali metal (from Cs to Li). The large energy gap between CBM and WF of X_2_CO_3_:MZO affects to electron transport ability due to the electron accumulation or blocking at the interface of EIL/ETL.

PL analysis of the active material on CTL is one of factors to estimate interface quality between CTL and EML. PL intensity of QDs on various CTLs can be affected by surface roughness and exciton decay time (τ). Figure 3a,b show the relative PL intensity and TRPL characteristics of R-QDs on X_2_CO_3_:MZO ETLs. It is noted that the standard PL intensity and exciton decay time are from R-QDs layer on glass substrate. The PL intensities of R-QDs on Rb_2_CO_3_:MZO, K_2_CO_3_:MZO and Na_2_CO_3_:MZO ETLs are similar, while the others showed a lower PL intensity by over 10% to that of standard R-QDs on glass. This can be explained as the surface morphology of X_2_CO_3_:MZO thin-films and will be further discussed with the AFM results. Figure 3b illustrates the TRPL characteristics of R-QDs on X_2_CO_3_:MZO ETLs and the details are summarized in Table 2. As shown in Figure 3b, the PL decay could be well fitted into a bi-exponential decay function, including short and long PL decays (τ_1_ and τ_2_) characterized as band-edge and trap exciton states, respectively [40]. Compared to TRPL of intrinsic R-QDs thin-film on glass, the trap exciton states (A_2_ on τ_2_) at the interface of X_2_CO_3_:MZO/R-QDs increase from 23.2% (on glass) to 26.2% (on Cs_2_CO_3_:MZO), 35.1% (on Rb_2_CO_3_:MZO) and to 42.1% (on K_2_CO_3_:MZO), and to 29.2% (on Na_2_CO_3_:MZO) and 26.8% (on Li_2_CO_3_:MZO), as shown in Table 2. The PL decay between band-edge (A_1_ on τ_1_) and trap (A_2_ on τ_2_) exciton states could be close to that of intrinsic R-QDs on glass substrate when Cs_2_CO_3_:MZO thin-film was used as ETL, therefore, we concluded that the Cs_2_CO_3_:MZO ETL helps to improve QLED performances. It is noted that the average exciton decay time (τ_avr_) of R-QDs on X_2_CO_3_:MZO thin-films was 20.0 ns on Cs_2_CO_3_:MZO, 17.0 ns on Rb_2_CO_3_:MZO, 16.8 ns on K_2_CO_3_:MZO, 17.7 ns on Na_2_CO_3_:MZO, and 18.5 ns on Li_2_CO_3_:MZO thin-films.

Figure 4 illustrates the AFM surface morphology of X_2_CO_3_:MZO ETLs on ITO substrate. The thickness of X_2_CO_3_:MZO thin-films was 60 nm and annealing conditions are 220 °C for 30 min in N_2_. It is confirmed that the peak-to-valley roughness (R_pv_) of X_2_CO_3_:MZO thin-films increases from 13.4 nm (Cs_2_CO_3_:MZO) to 55.7 nm (Rb_2_CO_3_:MZO) and to 17.8 nm for Li_2_CO_3_:MZO. This peak-to-valley roughness tendency is well matched to the PL intensity characteristics of R-QDs on X_2_CO_3_:MZO thin-films. The Cs_2_CO_3_:MZO thin-film exhibited the most smooth roughness with 13.4 nm for R_pv_, 1.5 nm for root-mean square roughness (R_q_) and 1.2 nm for average roughness (R_a_). More details are summarized in Table 3. The electrical properties of thin-film help to elucidate the charge transport in QLED. To study the electrical properties of X_2_CO_3_:MZO thin-films, the QD-free electron-only devices (EODs) were fabricated with the structure; ITO/LZO (60 nm)/X_2_CO_3_:MZO (60 nm)/LiF:Al. Figure 5a exhibits the energy band diagram of EODs resulted from UPS results in Figure 2. Through the previous UPS result, it is predicted that the electron transport ability of X_2_CO_3_:MZO degrades with decreasing atomic number of dopants. The current versus voltage characteristic of X_2_CO_3_:MZO thin-films are shown in Figure 5b. At the Ohmic contact region (J∝V, black dash line), it is confirmed that the conductivity of X_2_CO_3_:MZO decreases gradually with decreasing atomic number and the result is well matched to UPS result. The large energy barrier between CBM and WF hinders the efficient electron transport from cathode to QD EML, therefore, the reduction of conductivity can affect to charge balance in QD EML. The conductivities of X_2_CO_3_:MZO films are summarized in Table 4.

Capacitance versus voltage (C-V) is one of the most efficient methods to evaluate the charge accumulation at interface [41,42]. Figure 6 exhibits the capacitance of EODs with X_2_CO_3_:MZO ETL and QD EML. The energy band diagram of EODs is shown in Figure 6a and only energy barriers at the interfaces of LZO/X_2_CO_3_:MZO and X_2_CO_3_:MZO/QD can be considered in EOD because of the small energy barriers in LZO (CBM−WF = 0.11 eV) and R-QDs (CBM−WF = 0.08 eV) for the efficient electron injection from electrode. Figure 6b shows the capacitance versus frequency characteristics, measured with the alternating voltage (V_AC_) and applied DC voltage (V_DC_) were fixed as 100 mV and 0 V, respectively. It is confirmed that X_2_CO_3_:MZO based EODs (red, orange, light green, deep green, and blue symbols) have relatively higher capacitance than that of pristine MZO based EODs (black symbol). In Figure 6c, the capacitance versus voltage of EODs monitors the electron accumulation at interface of LZO/X_2_CO_3_:MZO under direct current (DC) bias. It is confirmed that the fast reduction of capacitance in Cs_2_CO_3_:MZO, Rb_2_CO_3_:MZO, and K_2_CO_3_:MZO ETL based EODs is generated with V_DC_ sweep from 0 to 4 V, while Na_2_CO_3_:MZO and Li_2_CO_3_:MZO based EODs exhibit the extremely slow reduction of capacitance compared to pristine MZO based EODs. It means that energy barrier for electron transport from LZO to R-QDs decreases by Cs_2_CO_3_, Rb_2_CO_3_ and K_2_CO_3_ blending in MZO, on the other hand, the energy barrier from LZO to R-QDs increases by Na_2_CO_3_ and Li_2_CO_3_ blending in MZO. The increased energy barrier by Na_2_CO_3_ or Li_2_CO_3_ blending in MZO hinders the electron transport from LZO to R-QDs, so that the electrons injected from cathode are accumulated at interface of LZO/X_2_CO_3_:MZO ETLs [43]. Therefore, the slow capacitance reduction is found. Figure 6d illustrates the electron accumulation mechanism resulted from C-V result and it can be explained as the energy barrier (CBM-WF) difference of X_2_CO_3_:MZO ETLs.

Electron accumulation at the interface tends to induce the electron trapping or charging in device, and it can be estimated by the hysteresis analysis with constant current stress as shown in Appendix A. The constant current of 50 mA was applied for 30 min in EODs and the thermal radiation was measured by infrared (IR) camera. It is noted that the black and red lines exhibit before and after constant current stress, respectively, and solid and dash lines exhibit positive and negative sweeps, respectively. Appendix A show the current versus voltage of EODs with MZO, Cs_2_CO_3_:MZO, Rb_2_CO_3_:MZO, K_2_CO_3_:MZO, Na_2_CO_3_:MZO, and Li_2_CO_3_:MZO ETLs (Inset: thermal radiation characteristic). In the inset images of Appendix A, the pristine MZO based EODs exhibited a higher thermal radiation of ~62 °C than that of X_2_CO_3_:MZO based EODs during constant current stress (36 °C for Cs_2_CO_3_:MZO, 39 °C for Rb_2_CO_3_:MZO, 40 °C for K_2_CO_3_:MZO, 41 °C for Na_2_CO_3_:MZO, and 43 °C for Li_2_CO_3_:MZO based EODs). The reduction of thermal radiation by introducing X_2_CO_3_ in MZO ETLs can be explained as the high thermal stability of Group 1-compounds based carbonates. The pristine MZO based EODs exhibited a high hysteresis with positive and negative sweeps, also, the resistance of MZO became higher after constant current stress (red line in Appendix A). Among the X_2_CO_3_:MZO ETLs, the Cs_2_CO_3_:MZO ETL based EOD showed a high electrical stability before and after constant current stress (between black dash and red solid lines). However, it is shown that the QLED with Cs_2_CO_3_:MZO based EOD had relatively large hysteresis before constant current stress than that with Rb_2_CO_3_:MZO or K_2_CO_3_:MZO. This relatively large hysteresis is related with high density of interface traps. The K_2_CO_3_:MZO ETL based EOD showed a smallest electron accumulation with positive and negative sweeps (between black solid and dash lines). However, similar behaviour of pristine MZO ETL based EODs can be seen and the resistance of K_2_CO_3_:MZO ETL based EOD increases slightly (black dash and red solid lines). On the other hands, the Na_2_CO_3_:MZO and Li_2_CO_3_:MZO ETLs still showed hysteresis characteristics after constant current stress, as shown in Appendix A (red solid and dash lines). These electrical hysteresis of ETL degrades QLED lifetime and more details are summarized in Appendix A. Therefore, we concluded that alkali metal carbonate blending in MZO affects to the electrical performance and stability of QLEDs.

### 3.3. Device Performance and Operational Lifetime of Inverted R-QLEDs with X_2_CO_3_:MZO ETLs

Figure 7 exhibits the device performance and lifetime of the inverted R-QLEDs with X_2_CO_3_:MZO ETLs. The energy band diagram of inverted R-QLEDs with X_2_CO_3_:MZO ETLs resulted from UPS data are illustrated in Figure 7a and device fabrication process is explained in Experimental Section. Figure 7b–e shows the current density and luminance characteristics as function of voltage (J-V and L-V) and current and power efficiencies as function of luminance (CE-L and PE-L). Compared to the X_2_CO_3_:MZO ETL based R-QLEDs, the pristine MZO ETL based one (grey line) exhibited the lower leakage and less forward currents, which can be explained by the high resistance of MZO as reported before [32]. In Figure 7b, we confirmed that the Cs_2_CO_3_:MZO ETL based R-QLED has relatively lower leakage currents because of the smoother roughness property of Cs_2_CO_3_:MZO thin-film than others. Also, the Li_2_CO_3_:MZO ETL based R-QLED exhibited the lowest current density at forward bias due to the inefficient electron transport of Li_2_CO_3_:MZO ETL from LZO to EML induced by the large energy barrier between CBM and WF. Inefficient electron transport hinders the efficient exciton generation in QD EML, so that Li_2_CO_3_:MZO ETL based R-QLED exhibited poor luminance and low efficiency as shown in Figure 7c. Figure 7d,e exhibit current and power efficiency of the inverted R-QLEDs with X_2_CO_3_:MZO ETLs, respectively. CE_max_ and maximum power efficiency (PE_max_) of inverted R-QLEDs are 16.3 cd/A and 20.5 lm/W with Na_2_CO_3_:MZO ETL, but the Na_2_CO_3_:MZO ETLs based R-QLED showed large CE roll-off phenomenon with increasing luminance. On the other hands, the Cs_2_CO_3_:MZO, Rb_2_CO_3_:MZO and K_2_CO_3_:MZO ETLs based R-QLEDs exhibited a slightly lower CE than that of Na_2_CO_3_:MZO ETLs based one, however, it exhibited the dramatically improved CE roll-off and bright luminance over 150,000 cd/m^2^. Figure 7f exhibits the operational lifetime of the inverted R-QLEDs with X_2_CO_3_:MZO ETLs with the initial luminance (L_0_) of 1000 cd/m^2^. For the QLED lifetime measurement, the encapsulated QLED samples are kept in dark box at room temperature with a humidity of ~40%. The lifetime characteristics of R-QLEDs showed a similar trend with CE roll-off phenomenon and the longest lifetime property was achieved for the inverted R-QLEDs with Rb_2_CO_3_:MZO ETLs as 620 h (@ T_95_). The Cs_2_CO_3_:MZO ETL based inverted R-QLEDs showed a competitive lifetime of 407 h (@ T_95_) compared to that of Rb_2_CO_3_:MZO ETL based device and detail performances are summarized in Table 5. Figure 7g illustrates the lifetime reduction mechanism of inverted R-QLEDs by introducing alkali metal carbonates into the MZO ETL. In this study, we found that the energy barrier, conductivity, interface and thermal stability of MZO could be controlled by the X_2_CO_3_ blending. Especially, we confirmed that the energy barrier in X_2_CO_3_:MZO ETLs become larger with decreasing atomic number of alkali metals in carbonate. The increased energy barrier in X_2_CO_3_:MZO ETL can generate electron accumulation or trap at interface of EIL/ETL and ETL/EML which degrade device lifetime of QLEDs. Compared to the stable hole transport from anode, the electron accumulation or trap at ETL interface hinder the efficient electron transport into QD EML, as a result, the device lifetime degrades. Therefore, we concluded that the Cs_2_CO_3_ and Rb_2_CO_3_ are efficient alkali metal carbonates in MZO ETL to improve device lifetime. The summarized lifetime and CE characteristics of inverted R-QLEDs reported in literatures are shown in Appendix A. It is noted that Cs_2_CO_3_:MZO and Rb_2_CO_3_:MZO ETLs based R-QLEDs exhibited the best performances in lifetime.

Figure 8 illustrates the summarized blending effect of alkali metal carbonates in MZO ETL for highly stable inverted R-QLEDs. It was found that the important factors to improve lifetime are energy barrier for efficient electron transport, T_g_ and conductivity of ETL. It is noted that high conductivity and low energy barrier (CBM-WF) of ETL induce a relatively low CE_max_ performance in R-QLEDs at low voltage, while it increases the lifetime of R-QLEDs along with improving the charge balance in R-QD EML at high applied voltage.

## 4. Conclusions

In this study, we report the blending effect of the various alkali metal carbonates in MZO ETL for highly stable R-QLEDs. Among the X_2_CO_3_ materials (X = Cs, Rb, K, Na, and Li), the inverted R-QLEDs with Cs_2_CO_3_:MZO, Rb_2_CO_3_:MZO, and K_2_CO_3_:MZO ETLs exhibited the improvement of lifetime over 100 times compared with that of R-QLEDs with MZO, Na_2_CO_3_:MZO, and K_2_CO_3_:MZO ETLs. The T_95_ is 407 h with Cs_2_CO_3_:MZO ETL, 620 h with Rb_2_CO_3_:MZO ETL, 94 h with K_2_CO_3_:MZO ETL, 0.07 h with MZO ETL, 0.2 h with Na_2_CO_3_:MZO ETL and 0.03 h with Li_2_CO_3_:MZO ETL. The X_2_CO_3_ blending concentrations in MZO were fixed as 4%. The blending effects of X_2_CO_3_ could be summarized as: (i) the T_g_ of electron transporting material increases by X_2_CO_3_ blending in MZO because of the less polarizing effect during its decomposition process; (ii) the conductivity increases by increasing atomic number (X_2_CO_3_ blending in MZO from Li to Cs. This improves the electron injection into QDs; (iii) the Fermi-levels of X_2_CO_3_:MZO become closed to conduction band with increasing atomic number (from Li to Cs); (iv) the charge accumulation at interfaces of the EIL/ETL and ETL/EML decreases with increasing atomic number (from Li to Cs) due to the reduction of energy barrier for electron transport, which improves lifetime of R-QLEDs.

## Figures and Tables

**Figure 1 nanomaterials-10-02423-f001:**
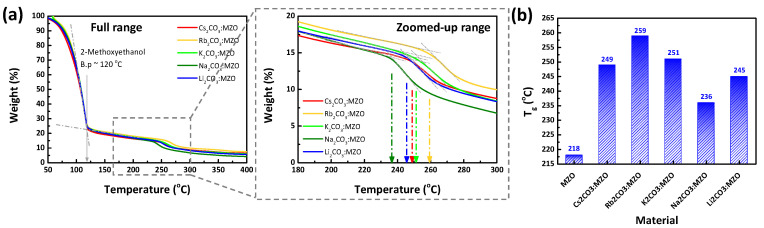
Thermogravimetric analysis (TGA) of alkali metal carbonate blended MZO (10% Mg doped Zinc Oxide) solutions. Weight loss characteristics of alkali metal carbonate blended MZO solutions with (**a**) full range (Inset: zoomed-up range). (**b**) The glass transition temperature (T_g_) measured for alkali metal carbonate blended MZO solutions.

**Figure 2 nanomaterials-10-02423-f002:**
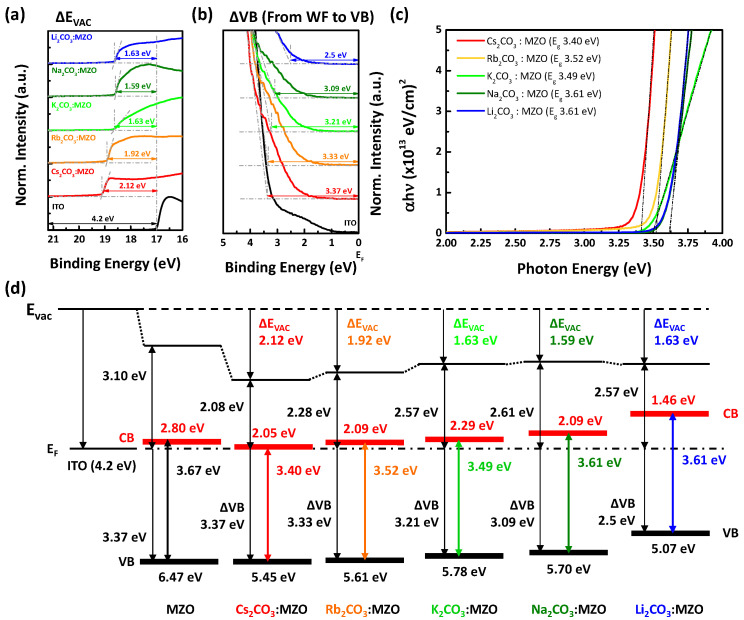
UPS results and energy band diagram of alkali metal carbonate blended MZO films. Ultraviolet photoelectron (UPS) spectra with (**a**) secondary electron cutoff and (**b**) valance band maximum (VBM) regions of alkali metal carbonate blended MZO thin films on ITO. (**c**) Tauc plot of alkali metal carbonate blended MZO thin films on glass with ~130 nm. (**d**) Energy band diagrams of ITO, MZO, Cs_2_CO_3_:MZO, Rb_2_CO_3_:MZO, K_2_CO_3_:MZO, Na_2_CO_3_:MZO and Li_2_CO_3_:MZO ETLs. The doping concentrations of alkali metal carbonates in MZO are fixed at 4 at%. It is noted that thick red and black solid lines in Figure 2d are CBM and VBM of the ETLs.

**Figure 3 nanomaterials-10-02423-f003:**
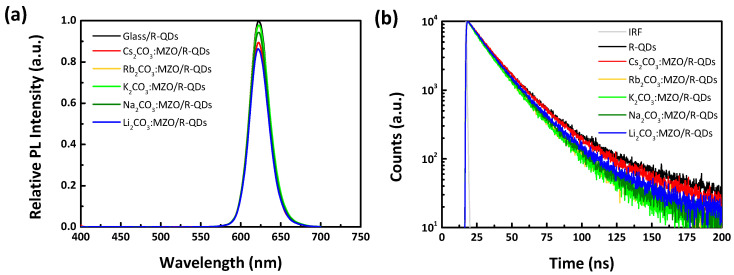
PL and its delay of R-QDs on alkali metal carbonate blended MZO films. (**a**) Relative PL intensity and (**b**) time-resolved PL (TRPL) characteristics of R-QDs on alkali metal carbonates blended MZO thin-films. Note that the standard PL intensity and exciton decay time are defined from R-QDs layer on glass substrate.

**Figure 4 nanomaterials-10-02423-f004:**
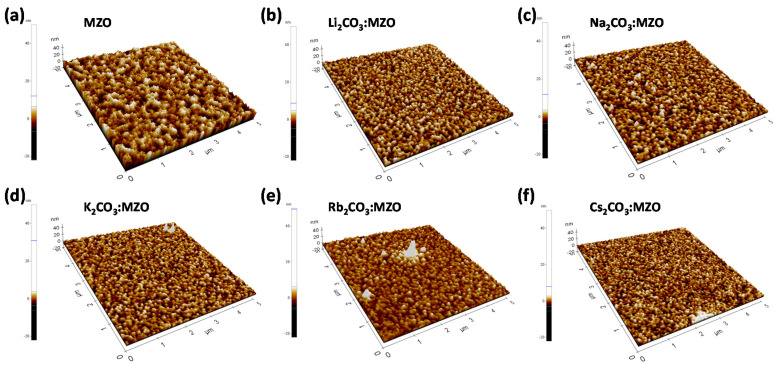
Atomic force microscopy (AFM) images of N_2_ annealed MZO and X_2_CO_3_:MZO thin-films (X = Li, Na, K, Rb and Cs) on ITO substrate. (**a**) MZO, (**b**) Li_2_CO_3_:MZO, (**c**) Na_2_CO_3_:MZO, (**d**) K_2_CO_3_:MZO, (**e**) Rb_2_CO_3_:MZO and (**f**) Cs_2_CO_3_:MZO thin-films. The thickness of thin-films is ~60 nm and it was deposited on ITO substrate.

**Figure 5 nanomaterials-10-02423-f005:**
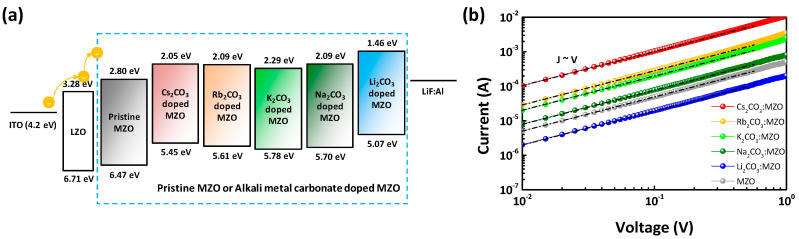
Current-voltage characteristics of pristine MZO and alkali metal carbonate blended MZO thin-films. (**a**) Energy band diagram and (**b**) current versus voltage characteristic of electron-only devices (EODs) except QD EML (Structure: ITO/LZO/MZO or X_2_CO_3_:MZO (X = Cs, Rb, K, Na and Li)/LiF:Al). Black dash-dot line exhibits the ohmic region of J∝V.

**Figure 6 nanomaterials-10-02423-f006:**
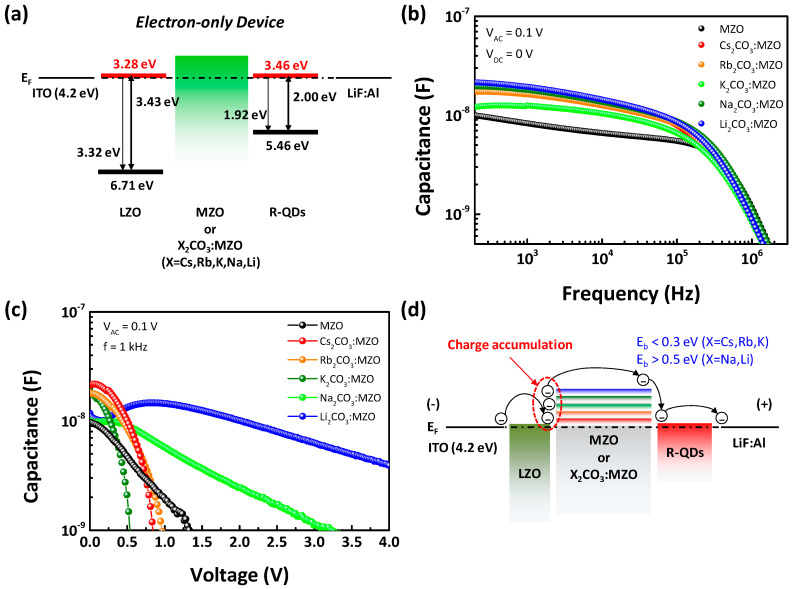
Capacitance and energy band of alkali metal carbonate blended MZO based electron-only devices (EODs) including QD EML. (**a**) Energy band diagram and (**b**–**d**) capacitance of EODs (Structure: ITO/LZO/X_2_CO_3_:MZO (X = Cs, Rb, K, Na, and Li)/QD/LiF:Al). (**b**) Capacitance versus frequency (C-f) and (**c**) capacitance versus voltage (C-V) of EODs. (**d**) Charge accumulation mechanism between LZO EIL and alkali metal carbonate blended MZO ETL from C-V characteristic.

**Figure 7 nanomaterials-10-02423-f007:**
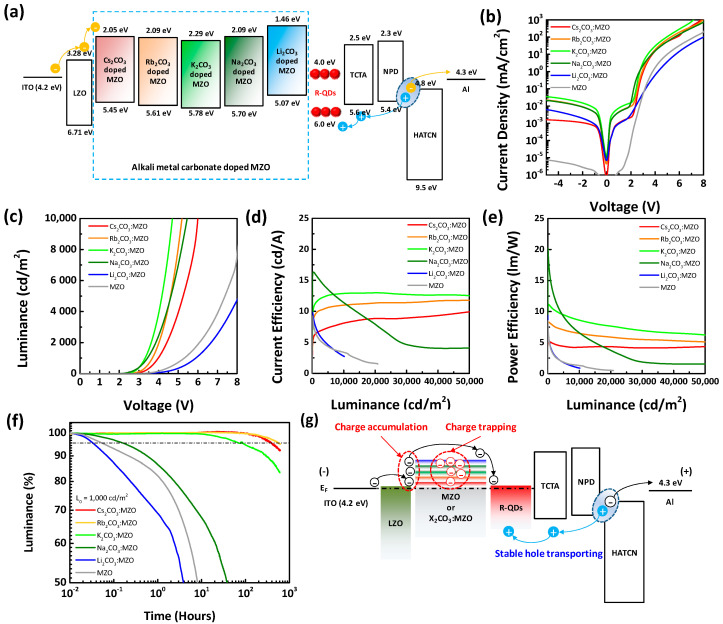
Device performance and lifetime of inverted R-QLEDs with pristine MZO and X_2_CO_3_:MZO ETLs. (**a**) Energy band diagram and (**b**–**e**) device performances of inverted R-QLEDs with pristine MZO and X_2_CO_3_:MZO ETLs. (**b**) Current density versus voltage, (**c**) luminance versus voltage, (**d**) current efficiency versus luminance, (**e**) power efficiency versus luminance, (**f**) operational lifetime characteristics, and (**g**) lifetime reduction mechanism of the inverted R-QLEDs with X_2_CO_3_:MZO ETLs.

**Figure 8 nanomaterials-10-02423-f008:**
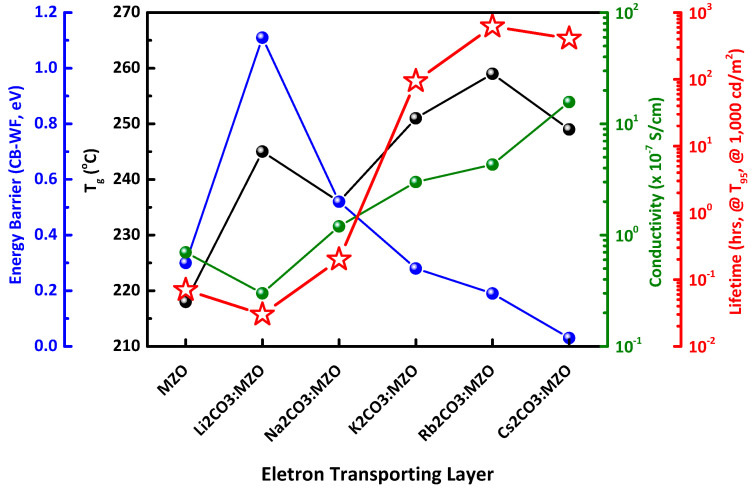
Summarized blending effect of alkali metal carbonates in MZO ETL for highly stable inverted R-QLEDs. Lifetime of inverted R-QLEDs with pristine MZO and X_2_CO_3_:MZO (X = Li, Na, K, Rb, and Cs) ETLs depends on thin-film characteristics (Energy barrier for electron transporting, T_g_ and conductivity characteristics).

**Table 1 nanomaterials-10-02423-t001:** The energy band characteristics of alkali metal carbonate blended MZO ETLs.

ETLs	Pristine MZO	Alkali Metal Carbonate Dopant in MZO				
		Cs_2_CO_3_	Rb_2_CO_3_	K_2_CO_3_	Na_2_CO_3_	Li_2_CO_3_
CB (eV)	2.80	2.05	2.09	2.29	2.09	1.46
WF (eV)	3.10	2.08	2.28	2.57	2.61	2.57
CB-WF (eV)	0.30	0.03	0.19	0.28	0.52	1.11
VB (eV)	6.47	5.45	5.61	5.78	5.70	5.07
E_g_ (eV)	3.67	3.40	3.52	3.49	3.61	3.61

**Table 2 nanomaterials-10-02423-t002:** Summarized exciton decay time of R-QDs on alkali metal carbonates blended MZO ETLs.

Exciton Decay Time	QD Underlayer					
	Glass	Alkali Metal Carbonate Blended MZO				
		Li_2_CO_3_	Na_2_CO_3_	K_2_CO_3_	Rb_2_CO_3_	Cs_2_CO_3_
τ_1_ (ns)	13.9	13.0	12.9	11.0	11.8	14.0
A_1_ (%)	76.8	73.2	70.8	57.9	64.9	73.8
τ_2_ (ns)	30.7	26.0	23.9	21.1	22.1	28.3
A_2_ (%)	23.2	26.8	29.2	42.1	35.1	26.2
τ_avr_ (ns)	20.6	18.5	17.7	16.8	17.0	20.0

**Table 3 nanomaterials-10-02423-t003:** Summarized AFM result of N_2_ annealed MZO and X_2_CO_3_:MZO thin-films (X = Li, Na, K, Rb and Cs) on ITO substrate.

Roughness	MZO	Alkali Metal Carbonate Blended MZO				
		Li_2_CO_3_	Na_2_CO_3_	K_2_CO_3_	Rb_2_CO_3_	Cs_2_CO_3_
R_pv_ (nm)	13.8	17.8	18.5	37	55.7	13.4
R_q_ (nm)	2.5	2.5	1.8	1.9	3.0	1.5
R_a_ (nm)	1.7	2.0	1.5	1.5	1.9	1.2

**Table 4 nanomaterials-10-02423-t004:** Summarized conductivity of pristine MZO and alkali metal carbonate blended MZO ETLs. The conductivity is calculated from the ohmic region.

Thin-Films	Pristine MZO	Alkali Metal Carbonate Blended MZO				
		Cs_2_CO_3_	Rb_2_CO_3_	K_2_CO_3_	Na_2_CO_3_	Li_2_CO_3_
σ (×10^−7^ S/cm)	0.7	15.7	4.3	3.0	1.2	0.3

**Table 5 nanomaterials-10-02423-t005:** Summarized device performance of inverted R-QLEDs with pristine MZO and X_2_CO_3_:MZO ETLs.

ETLs	V_T_ ^(1)^ (V)	V_D_ ^(1)^ (V)	CE_max_	PE_max_	L_max_	EQE_max_	@ 1k cd/m^2^		@ 10k cd/m^2^		@ 1k cd/m^2^
			(cd/A)	(lm/W)	(cd/m^2^)	(%)	CE (cd/A)	PE (lm/W)	CE (cd/A)	PE (lm/W)	T_95_ (h)
MZO	2.8	5.3	9.3	8.5	<30k	7.2	6.4	3.7	3.4	1.3	0.06
Li_2_CO_3_:MZO	2.5	5.9	9.9	9.4	<20k	7.4	7.6	4.0	2.8	0.9	0.03
Na_2_CO_3_:MZO	2.0	3.4	16.3	20.5	<70k	13.2	16.0	14.6	11.4	6.6	0.2
K_2_CO_3_:MZO	2.2	3.3	13.0	11.2	>150k	11.5	11.3	10.7	12.9	8.6	94
Rb_2_CO_3_:MZO	2.4	3.7	12.2	8.3	>150k	9.6	9.6	8.2	11.0	6.7	620
Cs_2_CO_3_:MZO	2.5	4.0	11.0	5.2	>150k	8.1	6.4	5.0	8.1	4.2	407

^(1)^ Turn-on and driving voltages (V_T_ and V_D_) are defined as the voltages when luminance are 1.0 cd/m^2^ and 1000 cd/m^2^, respectively.

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
