# Peer review of "Stability of Quantum-Dot Light Emitting Diodes with Alkali Metal Carbonates Blending in Mg Doped ZnO Electron Transport Layer"

_nanomaterials, 2020, doi:10.3390/nano10122423_

Round 1

Author Response

Best regards.

Hyo-Min Kim

Reviewer 2 Report

Review for the manuscript:

Entitled: " Stability of Quantum-dot Light Emitting Diodes with Alkali Metal Carbonates Doping in Mg Doped ZnO Electron Transport Layer"

for Nanomaterials.

With ID: nanomaterials-1004809

Dear authors,

Thank you for your manuscript.

General comments

Comments for the Authors

Occasionally you are asked to review manuscripts like this, but as hard as you try to find between the lines something to ask, at last you give up, since honestly, the article is flowless! This work is well within the scope of Nanomaterials and it may be of interest to most of the readers of this journal. It is well organized with good, references to follow. For all the above I have opted to recommend Acceptance for the manuscript in its current form.

Best regards

Author Response

Best regards.

Hyo-Min Kim

Reviewer 3 Report

The study deals with the doping by alkali metal carbonates of Mg doped ZnO electron transporting layer (ETL), to stabilize quantum dots base light emitting diodes (QLEDs). The systematic characterization of the ETL films indicates that the one doped with Cs2CO3 is the most stable and helps to improve QLED performances compare with others, above all regarding the PL and the roughness results. Moreover the energy barrier decreases in the presence of Cs2CO3. This leads to a high Luminance for the LEDs and an increased stability up to 407 hours. However, some important concerns need to be addressed:
1) The authors should clearly report in the Table 5 also the value of the EQE for each doping.
2) The thermal and electrical stability is well described, however it is not described how the stability of the devices is measured, in which conditions of ambient environment, and light, if the EQE is stable for all the time. 
3) The electrical hysteresis of Cs doped film needs more explanation, as it looks not in line with the conclusions.

Author Response

Best regards.

Hyo-Min Kim

Round 2

Reviewer 3 Report

 The manuscript is slightly improved compared to the original version, still some concepts need investigation in depth for future works, but the results are coherent with the conclusions, so it can be accepted in Nanomaterials.